# Highly Sensitive Magnetoelastic Biosensor for Alpha2-Macroglobulin Detection Based on MnFe_2_O_4_@chitosan/MWCNTs/PDMS Composite

**DOI:** 10.3390/mi14020401

**Published:** 2023-02-07

**Authors:** Xing Guo, Jianru Hou, Yang Ge, Dong Zhao, Shengbo Sang, Jianlong Ji

**Affiliations:** 1Shanxi Key Laboratory of Micro Nano Sensors & Artificial Intelligence Perception, College of Information and Computer, Taiyuan University of Technology, Taiyuan 030024, China; 2Key Lab of Advanced Transducers and Intelligent Control System of the Ministry of Education, Taiyuan University of Technology, Taiyuan 030024, China

**Keywords:** chitosan-coated MnFe_2_O_4_ nanoparticles, multi-walled carbon nanotubes, magnetoelastic effect, α2-M, magnetoelastic biosensor

## Abstract

The need for Alpha2-Macroglobulin (α2-M) detection has increased because it plays an important role in the diagnosis of diabetic nephropathy (DN). However, few sensors can realize the high-sensitive detection for α2-M with characteristics of being fast, flexible, wearable and portable. Herein, a biosensor based on a MnFe_2_O_4_@chitosan/MWCNTs/PDMS composite film was developed for α2-M detection. Due to the excellent magnetoelastic effect of MnFe_2_O_4_ nanoparticles, the stress signal of the biosensor surface induced by the specific antibody–antigen binding was transformed into the electrical and magnetic signal. Chitosan-coated MnFe_2_O_4_ particles were used to provide biological modification sites for the α2-M antibody, which simplified the conventional biological functionalization modification process. The MnFe_2_O_4_@chitosan particles were successfully prepared by a chemical coprecipitation method and the property was studied by TEM, FT-IR and XRD. MWCNTs were employed to enhance electrical conductivity and the sensitivity of the biosensor. The detection limit (LOD) was reduced to 0.1299 ng·mL^−1^ in the linear range from 10 ng∙mL^−1^ to 100 µg·mL^−1^, which was significantly lower than the limit of health diagnostics. The biosensor is fabricated by a simple method, with advantages of being rapid and highly-sensitive, and having selective detection of α2-M, which provides a novel method for the early diagnosis of DN, and it has potential in the point of care (PoC) field.

## 1. Introduction

Diabetic nephropathy (DN) is one of the most serious complications of diabetes; 30–50% of diabetes patients may develop DN [1,2]. In 2017, global diabetes nephropathy resulted in 348,959 deaths [3]. Early diagnosis facilitates the treatment of DN; therefore, the focus has shifted to discover and diagnose early DN. The key to the early diagnosis of DN is the highly-sensitive detection of disease markers.

In recent studies, researchers have proposed the following markers: (1) Neutrophil gelatinase-associated lipocalin (NGAL) is mainly stored in specific granules of neutrophils, but it also has low level of expression in other human tissues, so the specificity for DN detection is relatively low [4]. (2) The glomerular filtration rate (GFR) was measured by ^99m^Tc DTPA renal dynamic imaging [5]. This method needs injection into an elbow vein first, to be continuously monitored for 25 min, and then processed by software. This method is relatively time-consuming and labor-consuming. (3) Data also indicated that microRNA (miRNAs) is involved in the pathogenesis of DN. MiRNAs are highly conserved non-encoding RNAs oligonucleotide, but a previous study showed that miRNAs have little knowledge of the expression and function of DN [6]. 

In summary, the detection of these markers is costly, time-consuming and has low-specificity, so a sensitive and accurate biomarker is urgently needed to detect DN. At present, some laboratories have carried out research on using α2-M as a biomarker. α2-M is a protein in serum closely related to the development of DN. The concentration of α2-M in the serum of patients with DN is significantly higher than that of healthy people (1.52 ± 0.43 mg∙mL^−1^), so α2-M can be used as an important biomarker for the early diagnosis and prediction of DN. Several immunoassay techniques for α2-M have been developed, including single radial immunodiffusion (SRID), rocket immunoelectrophoresis (RIE), enzyme-linked immunosorbent assay (ELISA) and turbidimetric immunoassay (TIA) using polyethylene glycol (PEG). These methods for α2-M detection have some shortcomings, such as being time-consuming, having low accuracy and needing equipment or skills. Therefore, it is imperative to put forward a rapid and low-cost biosensor for α2-M detection.

With the rapid development of flexible electronic materials and sensing technology, flexible sensors have become more widely used in human health monitoring, biomedical science and flexible electronic skin [7]. Flexible sensors convert physiological activity signals into visual electrical signals in the form of signal conduction. Compared with traditional sensors based on metal and semiconductor materials, biosensors in this work have many advantages such as good flexibility, stretchability and continuous monitoring, which can meet the above requirements. Instant diagnosis has become a hot spot. Biosensors in this work are expected to be integrated in wearable health monitoring devices, and have huge market prospects in trillion level industries such as wise medicine and point of care testing (POCT). The signal conversion mechanism of a flexible sensor is mainly divided into three parts: piezoresistance, capacitance and piezoelectric [8]. The piezoelectric sensors are not suitable for static measurement. The piezoresistive sensor needs to adopt temperature compensation measures to keep its technical indicators such as zero drift and sensitivity at a high level. The capacitive pressure sensor is relatively difficult to process, and cannot isolate the measured gas or liquid. In the future, the research of flexible sensors will see new breakthroughs such as researching new functional materials, exploring new sensing mechanisms and developing new flexible sensor integration technology.

Active materials have been used to prepare biosensors with different response mechanisms, such as metal nanowires and metal nanoparticles. There is little research on ferrite materials as active materials, and ferrite materials have high resistivity, good dielectric properties, high permeability at high frequency and magnetoelastic effects compared with traditional materials. MnFe_2_O_4_ is a member of cubic spinel ferrite material, which has unique chemical properties [9] and is used widely in many fields. Furthermore, MnFe_2_O_4_ has excellent magnetoelastic effects, and can be used as active materials to manufacture biosensors with good performance. The magnetoelastic effect refers to the phenomenon that the magnetic properties are changed under the action of mechanical stress [10].

In this experiment, with the antibody on the biosensor surface combined with the antigen specifically, the stress (*σ*) caused by the combination led to the change in permeability (*µ*), thus causing the change in resistance (*R*). Based on this principle, the change in *σ* can be transformed into the change in *R*, and the detection of change in σ can be realized.

To improve the sensitivity, MWCNTs are used to construct the conductive network of biosensors and realize high-sensitive and rapid detection of α2-M. MWCNTs consist of multilayer graphite. When the α2-M antibody is specifically combined with the antigen, it produces immune complexes, which increase the stress on the biosensor’s surface. Then the gap between the MWCNT’s layers increase, leading to an increase in resistance. Therefore, the concentration of the α2-M antigen can be reflected by the change in resistance before and after antigen detection.

Here, a biosensor based on MnFe_2_O_4_@chitosan/MWCNTs/PDMS for α2-M detection is presented for the first time. The successful preparation of MnFe_2_O_4_@chitosan by a chemical coprecipitation method provides sites for the highly efficient binding of the α2-M antibody. The relationship between the resistance change and the concentration of α2-M is established, and the hysteresis loops measurement of the biosensor before and after the α2-M antigen detection are tested. Meanwhile, the performance of the biosensor is evaluated. The experimental results show excellent performance of the biosensor for α2-M detection, with advantages of a simple preparation method, low-cost, single-use, high sensitivity, strong specificity, a large linear range and low LOD. Because the biosensor has the advantages of convenience and high sensitivity, it has the potential to detect diseases quickly and accurately. The development of the biosensor has also contributed to the field of point of care.

## 2. Materials and Methods 

### 2.1. Chemicals and Materials

PDMS was purchased from Zhuhai kailibang Technology Co., Ltd. (Zhuhai, China) α2-M antibody and α2-M antigen were purchased from Shanghai Haling Biotechnology Co., Ltd. (Shanghai, China) 1-ethyl-3-(3-dimethylaminopropyl) carbodiimide hydrochloride (EDC, CAS: 25952-53-8), N-hydroxysulfosuccinimide (NHS, CAS: 6066-82-6), phosphate buffered saline (PBS, 0.01 M, pH = 7.4) and bovine serum albumin (BSA, 0.1%, CAS: 9048-46-8) were purchased from Sigma-Aldrich Corporation (Saint Louis, MO, USA). MWCNTs has a diameter of 4–6 nm, length of 10–20 μm and purity of >98 wt% (Chengdu Organic Chemistry Co., Ltd., Chengdu, China, Chinese academy of sciences zhongke times nanometer). The water used in the experiments was obtained from an ultra-pure water manufacturing system (URT-11-10T).

Digital multimeter (Keithley 2400, Tektronix, Shanghai, China, CHN) was used to measure the electrical properties of the biosensors. Scanning electron microscope (SEM, GeminiSEM 300, GER) was employed to observe the surface topographies of the biosensors. Energy dispersive spectrometer (EDS) was used to detect the elements of the biosensor’s surface. Transmission electron microscopy (TEM, Thermo Scientific, Waltham, MA, USA, Talos F200X) was used for TEM imaging. X-ray diffraction (XRD) patterns were measured to obtain information on the phases and crystalline structures. Integrated physical property measurement system (PPMS, QUANTUM Scientific Instrument Trading (Beijing) Co., Ltd., Beijing, China) was used to measure hysteresis loop.

### 2.2. Preparation of Chitosan-Coated MnFe_2_O_4_ (MnFe_2_O_4_@chitosan)

According to the method in the literature [11], the preparation process of MnFe_2_O_4_@chitosan is exhibited in Figure 1. Firstly, 0.8 g MnFe_2_O_4_ was dispersed in 30 mL of chitosan solution (4 mg∙mL^−1^) dissolving in 2% acetic acid solution. The suspension was mixed by sonication for 30 min, and then 15 mL of sodium tripolyphosphate solution (0.5 mg∙mL^−1^) added into the above solution, then followed by persistent stirring for 30 min. The MnFe_2_O_4_@chitosan were collected from the solution and washed by water and ethanol 3 times. Finally, the products were dried at 50 ℃ for 1 h in a drying oven.

### 2.3. Preparation of MWCNTs and Fabrication of the MnFe_2_O_4_@chitosan/MWCNTs/PDMS Composite Film

MWCNTs were used to enhance the electrical conductivity and sensitivity of biosensor. Firstly, MWCNTs were prepared. To do this, take 50 mg of MWCNTs into a container, then add 5 mL deionized water, and then use ultrasonic cell crusher for good dispersion for 15 min.

The fabrication process of the MnFe_2_O_4_@chitosan/MWCNTs/PDMS composite film based on PDMS is shown in Figure 2. Mix PDMS prepolymer and curing agent with a 10:1 wt ratio and then remove bubbles. Then add the prepared MnFe_2_O_4_@chitosan and MWCNTs into the mixture with 25:3:100 wt ratio. The composite film (4 mm × 4 mm × 0.08 mm) was fabricated by spin coater and dried on drying table at 60 ℃ for 1 h. Finally, the composite film was cleaned by alcohol and deionized water, respectively, 3 times and dried at 60 ℃ for 30 min.

### 2.4. Antibody Immobilization

The process of antibody immobilization is shown in Figure 3. The α2-M antibody was diluted into PBS buffer to prepare 50 µg·mL−1 antibody solution. The α2-M antibody was activated using the 4 mg∙mL−1 EDC/NHS at 37 ℃ for 0.5 h ,which made the bonding with the amino group on the CS more efficient. The biosensors were immersed into the antibody solution at 37 ℃ for 1 h so that the antibodies could be efficiently immobilized on biosensors. Then, the antibody-modified biosensors were rinsed with PBS to remove the antibodies adsorbed physically. Next, the biosensors were washed by BSA (0.1%) for 0.5 h to block the physically adsorbed sites. Then, the biosensors were rinsed by PBS, then dried to obtain the high-sensitive magnetoelastic biosensor based on MnFe_2_O_4_@chitosan/MWCNTs/PDMS composite for α2-M detection.

## 3. Results and Discussion

### 3.1. Characterization of the MnFe_2_O_4_@chitosan

The size and shape of the naked MnFe_2_O_4_ and MnFe_2_O_4_@chitosan are characterized by TEM in Figure 4. The naked MnFe_2_O_4_ presented a regular size with a diameter of about 31 nm (Figure 4a). After being coated with chitosan (Figure 4b), MnFe_2_O_4_@chitosan had a typical core-shell structure with better dispersion, and the diameter was about 46 nm. The increase in particle diameter before and after coating proves the coating is successful.

Figure 5a shows the FT-IR spectra of the MnFe_2_O_4_ and MnFe_2_O_4_@chitosan. The peaks around 576 cm^−1^ in the curve correspond to the Fe-O group. Compared to MnFe_2_O_4_, the FT-IR spectra of MnFe_2_O_4_@chitosan show new peaks occurred at 1234 cm^−1^ and 1059 cm^−1^, which were related to the C-N group and C-O-C group in the chitosan. The appearance of new peaks proves that chitosan was successfully coated on MnFe_2_O_4_. Other peaks had little change. XRD assays of the MnFe_2_O_4_ and MnFe_2_O_4_@chitosan are shown in Figure 5b. The MnFe_2_O_4_ and MnFe_2_O_4_@chitosan at 30.1°, 35.4°, 43.1°, 53.4°, 57.0° and 62.6° both appeared at diffraction peaks, which were (220), (311), (400), (422), (511) and (440), respectively. These peaks have good correspondence to the MnFe_2_O_4_ standard cards JCPDS73-1964 (lattice constant: a = b = c = 8.515 A, space group: FD-3m), indicating the MnFe_2_O_4_@chitosan has a spinel structure and CS has no effect on the crystallinity of MnFe_2_O_4_. Therefore, the preparation of MnFe_2_O_4_@chitosan was proven to be successful by FT-IR and XRD.

### 3.2. Characterization of the MnFe_2_O_4_@chitosan/MWCNTs/PDMS Flexible Biosensor

Figure 6 shows the stress–strain curve of the biosensor; the Young's modulus of the biosensor can be calculated as 3.0136 Mpa by the formula: *E* = *σ*/*ε*, where *E* is the Young's modulus, *σ* is stress and *ε* is strain. The Young's modulus is only related to the physical properties of the material, reflecting the stiffness of the material. The higher the Young's modulus is, the less likely the material will be deformed. The MnFe_2_O_4_@chitosan/MWCNTs/PDMS flexible biosensor has low Young's modulus, which shows the biosensor has good performance in stretchability.

SEM and EDS were assessed in the process of experiments as shown in Figure 7. Figure 7a shows that MnFe_2_O_4_@chitosan and MWCNTs were uniformly dispersed on the surface of the biosensor, and the biosensor surface appeared to intricately overlap a porous fiber network composed of spherical MnFe_2_O_4_@chitosan and reticular MWCNTs. The aggregates of the macromolecules of the α2-M antibody can be observed in Figure 7b. Compared to Figure 7b, the macromolecular immunocomplex on the surface of the biosensor in Figure 7c became more prominent, which indicates the antigen was successfully bound to the antibody.

For further demonstration of the surface modification process of the biosensors, the EDS characterization of the biosensor's surfaces during the preparation and detection process were measured, revealing the existence of C, N, O, Fe, Mn and Si in the MnFe_2_O_4_ and PDMS substrate (Figure 7g–i). To see the changes in the content of each element more clearly, the EDS analysis is shown in Figure 7d–f. Figure 7d,g show the EDS analysis and EDS images of the biosensor surface. Figure 7e,h show the EDS analysis and EDS images of the surface of the biosensor modified with the antibody. From Figure 7d,g, the amount of C, N and O in the antibody-modified biosensor is higher than that in the non-antibody-modified biosensor, which indicates that the bio-sensor was modified with the antibody successfully. Figure 7f,i show the EDS analysis and EDS images of the biosensor surface that detected α2-M. After detecting the α2-M antigen, the amounts of C, N and O were even higher than those in that only modified with the antibody because the immune complex caused by the antigen binding with the antibody belongs to protein molecules, consisting of C, O and N. No matter whether the biosensor is modified by the antibody or not, the counts of Fe, Mn and Si remained unchanged because the elements that compose the antibody do not contain these elements. Figure 7 indicated that MnFe_2_O_4_@chitosan, the antibody and the antigen were successfully deposited on the MnFe_2_O_4_@chitosan/MWCNTs/PDMS biosensor.

### 3.3. Optimization of the Concentration of α2-M Antibody

To improve the performance of the biosensors, the concentration of the α2-M antibody optimization experiment was carried out and repeated three times. Under the same concentration of the antigen, four different concentrations (25, 50, 100, 200 μg∙mL^−1^) of the antibody were detected then the resistance change (Δ*R*) was calculated. Figure 8 shows the Δ*R* of the biosensor modified with different concentrations of the antibody. From Figure 8, we can see that the Δ*R* is most obvious when the concentration of the antibody is 50 μg∙mL^−1^. When the modification concentration is greater than 50 μg∙mL^−1^, the carboxyl group of antibodies will competitively bind to the amino group on the surface of MnFe_2_O_4_@chitosan, resulting in the antibody not fully fixing on the surface of the biosensor. When the antibody modification concentration is less than 50 μg∙mL^−1^, the antibody is not wholly modified on the surface of MnFe_2_O_4_@chitosan. Hence, when the concentration of the antibody modification is 50 μg∙mL^−1^, the α2-M antibody can be adequately modified on the biosensor’s surface, and the detection sensitivity is at its highest. Thus, the optimal concentration of the modification is 50 μg∙mL^−1^.

### 3.4. α2-M Detection

Figure 9 shows the relationship between the ΔR and the logarithm of α2-M antigen concentration. Illustration (a) is a schematic diagram of the process of the antigen binding with the antibody and illustration (b) is a picture of the experiment process. The α2-M antigen combined with the antibody modified on the biosensor produced an immune complex, which increased the lamellar spacing of MWCNTs with a porous structure, causing the contact resistance of the biosensors to increase. In Figure 9, it can be seen that the ΔR increased with the increase in α2-M antigen concentration. By linear fitting, the functional relationship between the logarithm of α2-M antigen concentration and the Δ*R* was obtained: ΔR = 0.2983 lgC_α2-M_ + 0.33806, with the linear correlation coefficient *R*^2^ = 0.99629. Three experiments were repeated under the same conditions, which proved that the sensor has good repeatability. Consequently, the linear range of the biosensor for α2-M detection is 10 ng∙mL^−1^ – 100 µg·mL^−1^ , and the detection limit is 0.1299 ng∙mL^−1^.

The comparison of the biosensors in this work and other methods previously reported is shown in Table 1. The biosensor is simple to manufacture, low in cost and easy to operate. After detecting α2-M, the biosensor obtained a wider linear range and a lower detection limit. From Table 1, the performance of the biosensor in this work is even better than other methods, with a wider linear range, higher accuracy and the lowest cost. Hence, the biosensors exhibit the potential for detecting α2-M.

### 3.5. Hysteresis Loop Measurement of the Biosensor

Figure 10 shows a hysteresis loops measurement of the biosensor before and after the detection of the α2-M antigen (100 μg∙mL^−1^) at 300 K. We placed the sample in the sample rod, and then placed the sample rod vertically in the bin. The biosensor was positioned horizontally with respect to the magnetic field main axis. As shown in Figure 10, under the same magnetic field intensity (*H*), the biosensor’s magnetization (*M*) after α2-M detection was higher than before α2-M antigen detection. According to the equation of *M* = *X* × *H*, the direct current (DC) magnetic susceptibility (X) of the biosensor was increased. According to the relationship between μ and X: *μ* = 1 + *X*. After the antigen detection, μ of the biosensor was also increased. When MnFe_2_O_4_@chitosan modified with the antibody was combined with the antigen specifically, the magnetization state of MnFe_2_O_4_@chitosan changed, which caused the increase in the relative permeability and DC susceptibility of the biosensor.

### 3.6. Specificity Measurement

To confirm the specificity of the biosensor, α2-M, human serum albumin (HSA), c-reactive protein (CRP), uric acid, African swine fever virus (ASFV) and bovine serum albumin (BSA) with the same concentrations (100 μg∙mL^−1^) were detected and this was repeated three times for each substance. The responses of the biosensor are shown in Figure 11. When testing α2-M, the biosensor had a more obvious response than other substances, indicating that the biosensor has high specificity for α2-M detection. The reason for this phenomenon is that the specific binding of the antibodies and antigens caused surface stress on the biosensor. In addition, the antigens of other biological molecules did not bind to the α2-M antibody, which led to a weak response.

## 4. Conclusions

In conclusion, the MnFe_2_O_4_@chitosan/MWCNTs/PDMS biosensor was firstly reported for high-sensitive detection of α2-M. The MnFe_2_O_4_@chitosan nanoparticles were employed to magnify the membrane deformation to realize the conversion of surface stress to the electrical and magnetic signal. The MnFe_2_O_4_@chitosan/MWCNTs/PDMS biosensor was fabricated using a low-cost method, and EDS, SEM, TEM, XRD and FT-IR measurement were used to confirm that the fabrication of MnFe_2_O_4_@chitosan and the biosensor were successful. The stress caused by the specific binding of the antibody and antigen caused an increase in µ, thus causing a change in R. Therefore, the biosensor realized the signal conversion. The biosensor detected α2-M with a wide linear range from 10 ng∙mL^−1^ to 100 μg∙mL^−1^, and an LOD as low as 0.1299 ng∙mL^−1^, which is lower than the limit of health diagnostics (1.52 ± 0.43 mg∙mL^−1^). Furthermore, the biosensor possessed the potential for the diagnosis of early DN with the advantage of portability and high sensitivity.

## Figures and Tables

**Figure 1 micromachines-14-00401-f001:**
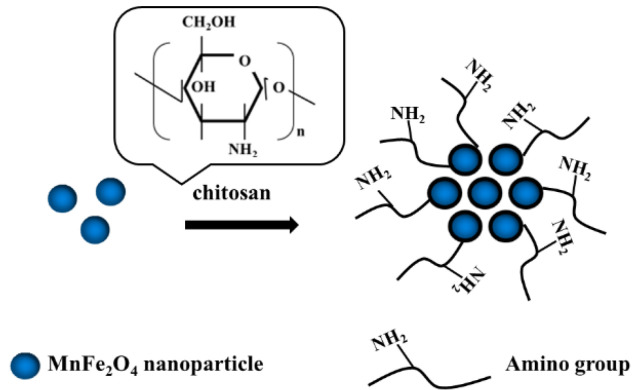
Schematic diagram for fabrication of MnFe_2_O_4_@chitosan.

**Figure 2 micromachines-14-00401-f002:**
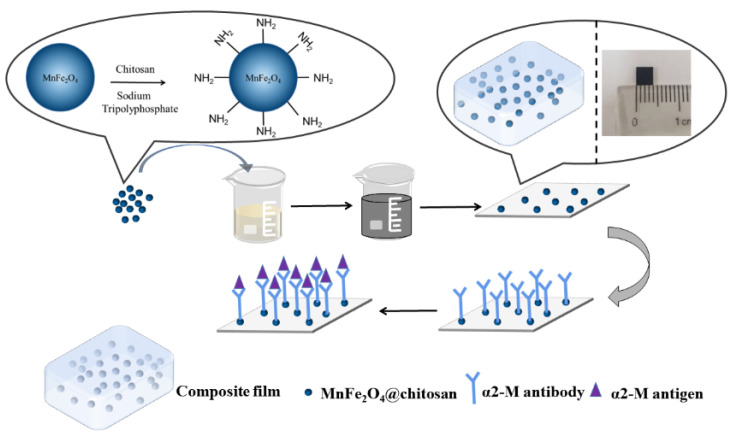
Schematic diagram for fabrication of MnFe_2_O_4_@chitosan/MWCNTs/PDMS composite film.

**Figure 3 micromachines-14-00401-f003:**
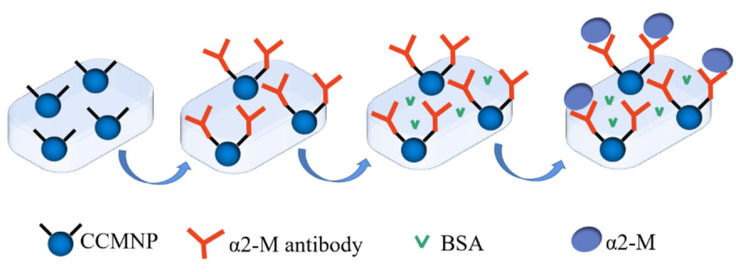
Modification process of α2-M antibody on the biosensor.

**Figure 4 micromachines-14-00401-f004:**
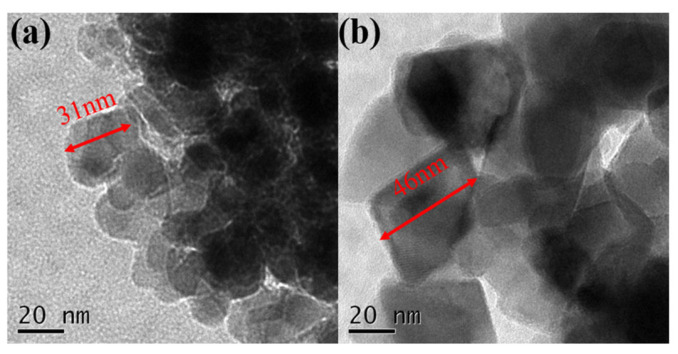
TEM images of naked MnFe_2_O_4_ (**a**) and MnFe_2_O_4_@chitosan (**b**).

**Figure 5 micromachines-14-00401-f005:**
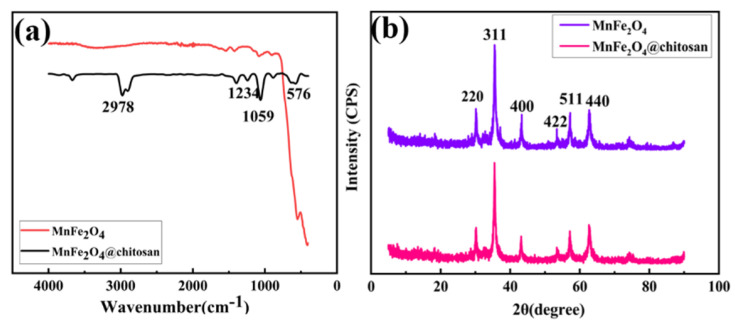
FT-IR spectra (**a**) and XRD (**b**) of naked MnFe_2_O_4_ and MnFe_2_O_4_@chitosan.

**Figure 6 micromachines-14-00401-f006:**
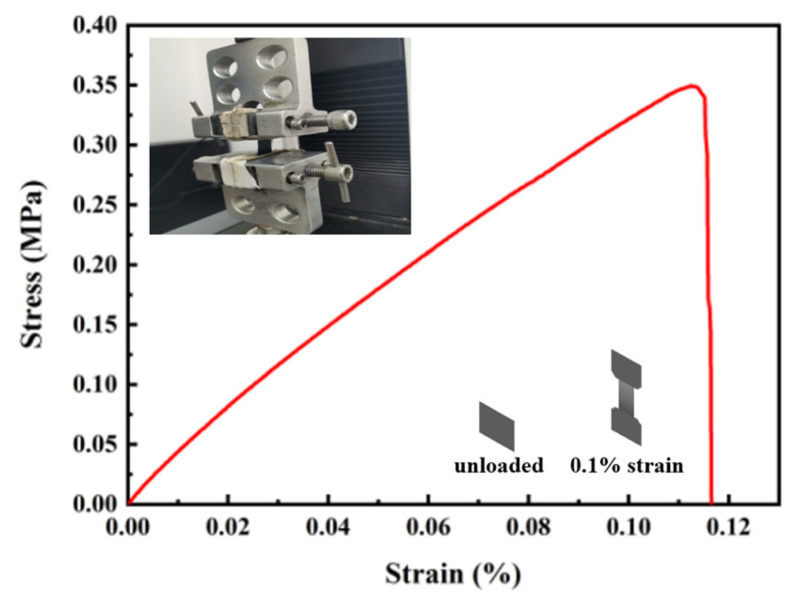
Stress–strain curve of MnFe_2_O_4_@chitosan/MWCNTs/PDMS biosensor.

**Figure 7 micromachines-14-00401-f007:**
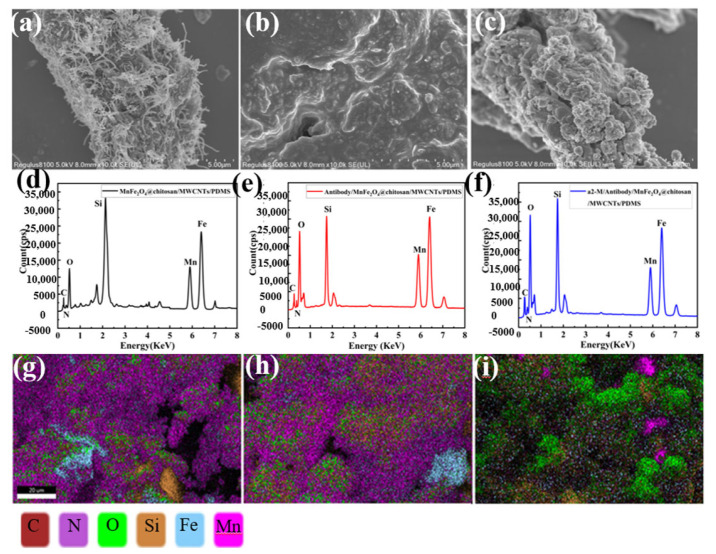
SEM (**a**) and EDS analysis (**d**) and EDS images (**g**) of the MnFe_2_O_4_@chitosan/MWCNTs/PDMS biosensor surface; SEM (**b**) and EDS analysis (**e**) and EDS images (**h)** of the MnFe_2_O_4_@chitosan/MWCNTs/PDMS/antibody; SEM (**c**) and EDS analysis (**f**) and EDS images (**i**) of the MnFe_2_O_4_@chitosan/MWCNTs/PDMS/antibody/α2-M antigen biosensor surface.

**Figure 8 micromachines-14-00401-f008:**
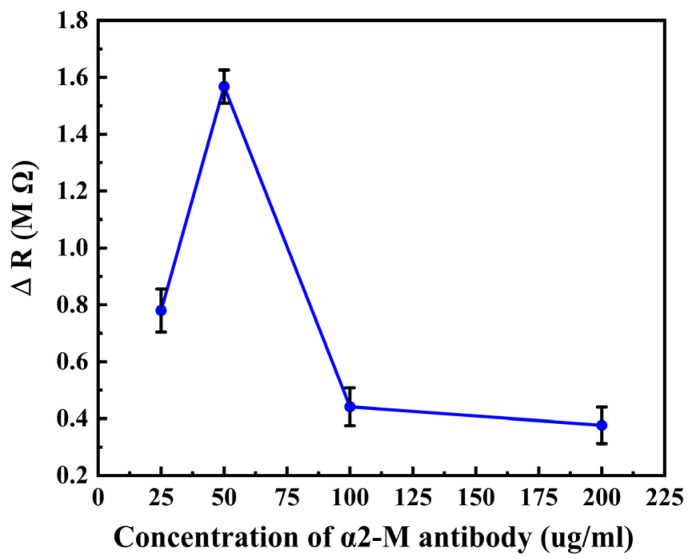
Δ*R* corresponding to different concentrations of α2-M antibody.

**Figure 9 micromachines-14-00401-f009:**
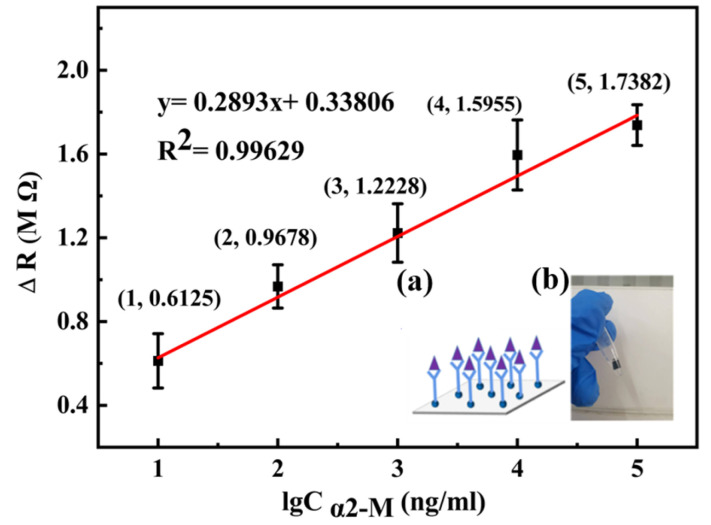
∆*R* of the biosensor detecting α2-M in the range of 10 ng∙mL^−1^–100 µg∙mL^−1^ and linear fitting curve. Illustration (**a**) is a schematic diagram of the process of the antigen binding with the antibody and illustration (**b**) is a picture of the experiment process.

**Figure 10 micromachines-14-00401-f010:**
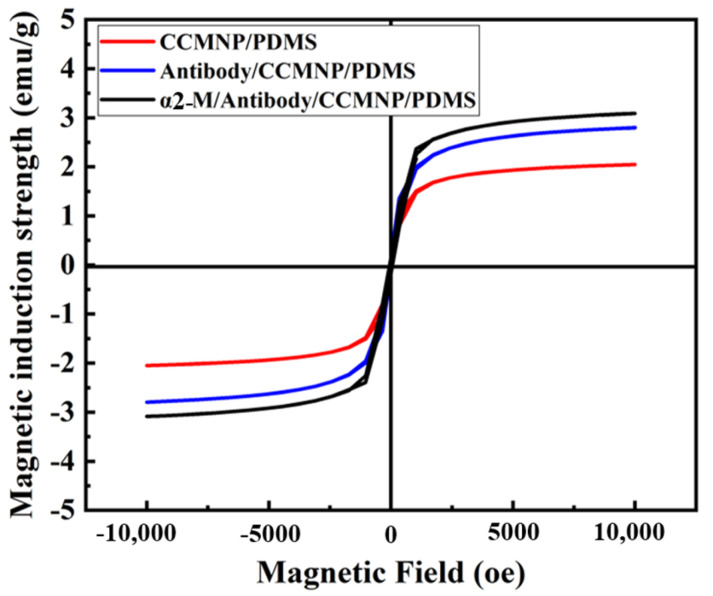
Hysteresis loops measurement before and after antigen detection.

**Figure 11 micromachines-14-00401-f011:**
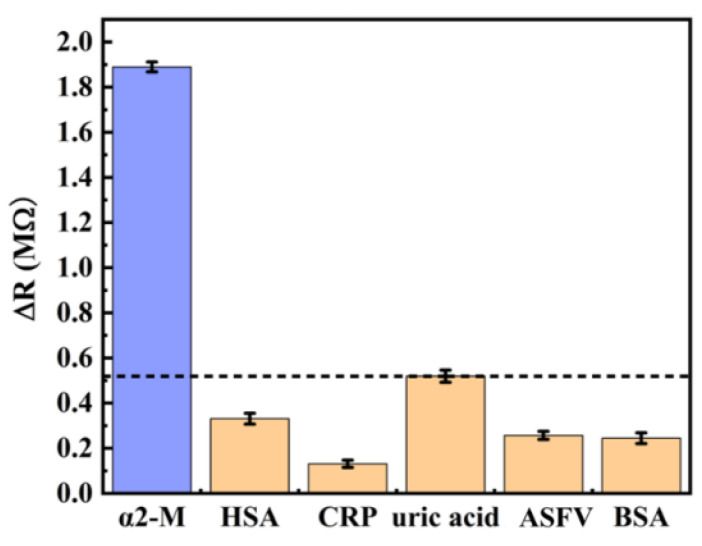
Specificity measurement of the biosensor.

**Table 1 micromachines-14-00401-t001:** Comparisons of performances between various methods for α2-M detection.

Detection Method	Linear Range (μg∙mL^−1^)	LOD (μg∙mL^−1^)	Assay Time	Ease of Use	Ref.
Quantitative immunoassay	2–1000	3	Several hours	Low-accuracy	[12]
Turbidimetric immunoassay	120–5000	120	Several minutes	Needs skill	[13]
ME biosensor based on MnFe_2_O_4_@chitosan/MWCNTs/PDMS	0.01–100	1.299 × 10^−4^	Several minutes	Minimum skill; smaller size	This work

## Data Availability

The data that support the findings of this study are available upon reasonable request from the authors.

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
