# Peer review of "Highly Sensitive Magnetoelastic Biosensor for Alpha2-Macroglobulin Detection Based on MnFe2O4@chitosan/MWCNTs/PDMS Composite"

_micromachines, 2023, doi:10.3390/mi14020401_

Round 1
Reviewer 1 Report
This study (micromachines-2113046) established a flexible magnetoelastic biosensor for Alpha2-Macroglobulin detection with MnFe2O4@chitosan/MWCNTs/PDMS composite film. The characteristics of materials have been validated, but there is still a lack of sensor signal description, analysis and discussion of the results. As a result, major revision of this manuscript is suitable. The points of revisions are listed below:
1. The necessity of flexible sensing is not made clear in this paper. Specific research should be carried out according to detection requirements. If the marker is present in the blood, then the author should clarify whether the substance requires long-term monitoring. Is it suitable for flexible sensing system detection? These questions need further clarification and answers.
2. Magnetic sensors are used for detection. What is the base of the sensors not described in the paper? To which substrate is the magnetic material immobilized? What is the detection signal? In addition, the original signal of sensor detection is also missing in the paper.
3. For the Stress-Strain section (Figure 6), please provide the original tensile picture of the material.
4. According to Figure 6, is the sensor made of flexible material as a whole?
5. Please further check the correctness of the SEM image. In particular, Figure 7b is quite different from other figures. In theory, Figure 7b and Figure (a)(c), although different in details, will be similar in overall form.
6. Provide the original signals in Figure 9 and place it in the manuscript. In addition, the 2 illustrations in the Figure 9 are not mentioned in the text, please delete or add descriptions.
Reviewer 2 Report
The authors present a flexible biosensor based on MnFe2O4@chitosan/MWCNTs/PDMS composite film for Alpha2-Macroglobulin detection. This biosensor is fabricated by a simple method, which provides a novel method for early diagnosis of Diabetic nephropathy.
I have the following comments:
- There are many English mistakes; authors should go all over the text and correct them
- In section 3.1 (Characterization of the MnFe2O4@chitosan) does not make physical sense significant digits 0.1nm in 30.9 nm and 46.3 nm. How the measurements where done? What is the error bar in those measurements?
- In section 3.3. (Optimization of the concentration of α2-M antibody), there is no information at all about the experiment to determine the resistance change (ΔR). How was the experimental setup? Also, Figure 8, showing the values of ΔR corresponding to different concentrations of α2-M antibody is absolutely unacceptable. There are no error bars and to conclude - with just FOUR experimental data - that the optimal concentration of the modification is 50 μg∙ml−1 is also unacceptable.
- In section 3.5. (Hysteresis loop measurement of the biosensor) there are more problems. Authors MUST say the temperature of hysteresis measurements in the text and also in the figure. Also, they must clearly indicate the geometry between sample and coils measurements system. The sample was positioned vertically or horizontally respect to the magnetic field main axis? Did the authors take into account demagnetization effects? Clearly the authors are not form the area of magnetism. They have to be more formal with those concepts. For example, the equation (and NOT the formula) defining M is not that one. Are they talking about AC or DC magnetic susceptibility? Obviously is the DC one. However, in science nothing has to be obvious but clear. Magnetic measurements are NOT that simple. What is the physical meaning of the shape of the MxH curve? That is a superparamagnet, so which are de biological implications of that behavior? What caused the increase of relative permeability and susceptibility of the biosensor?
The authors should go all over those comments before the manuscript could be accepted.
Round 2
Reviewer 1 Report
The author has answered all the questions. The article can be accepted.
Author Response
We thank the reviewer for accepting our paper.
Reviewer 2 Report
The authors have satisfactorily done all required corrections and changes.
The manuscript can now be accepted in this form.
Author Response

(The authors gave the same response as above.)
